# Comparative microfluidic and enzymatic analyses reveal multifaceted snake venom resistance and novel VWF behavior in the opossum *Monodelphis domestica*

**Matthew L. Holding** [1,2*☉], **Dante Disharoon** [3☉], **Laura M. Haynes** [1☉],
**Bipin Chakravarthy Paruchuri** [3], **M. Hao Hao Pontius** [4], **Krista Golden** [4],
**Jordan A. Shavit** [4,5], **Karl C. Desch** [4], **David Ginsburg** [1,4,5,6], **Anirban Sen Gupta** [3],
**Yolanda Cruz** [7], **Danielle H. Drabeck** [8*]

**1** Life Sciences Institute, University of Michigan, Ann Arbor, Michigan, United States of America,
**2** Department of Ecology and Evolutionary Biology, University of Michigan, Ann Arbor, Michigan,
United States of America, **3** Department of Biomedical Engineering, Case Western Reserve University,
Cleveland, Ohio, United States of America, **4** Department of Pediatrics, University of Michigan, Ann Arbor,
Michigan, United States of America, **5** Department of Human Genetics, University of Michigan, Ann Arbor,
Michigan, United States of America, **6** Department of Internal Medicine, University of Michigan, Ann Arbor,
Michigan, United States of America, **7** Department of Biology, Oberlin College, Oberlin, Ohio, United
States of America, **8** Department of Ecology, Evolution, and Behavior, University of Minnesota, Twin Cities,
Minnesota, United States of America

☉ These authors contributed equally to this work.
* matthewholding28@gmail.com (MLH); danielle.drabeck@gmail.com (DHD)

## Abstract

Interactions between predators and prey are often characterized by strong selection pressures that shape extreme physiological adaptations. Venom resistance in large-bodied South American opossums (Clade Didelphini) is a striking example, as these marsupials prey on venomous snakes and exhibit remarkable resistance to their venom. While resistance is well documented in Didelphini, relatively little is known about venom resistance in the smaller, more diverse members of Didelphidae, which inhabit the same regions and encounter the same predators. Moreover, resistance of opossum von Willebrand factor (VWF) to the venom C-type lectin-like proteins has not been previously studied under simulated vascular flow states. Here, we use microfluidic devices to investigate venom resistance in the small-bodied opossum, *Monodelphis domestica,* examining platelet adhesion and fibrin deposition in response to purified venom components. Additionally, we conduct platelet aggregometry and assays of serum protease inhibitors in the presence of venom from sympatric and allopatric vipers to examine patterns of species-specificity and adaptation. Our results show that *M. domestica* resists venom-induced disruptions to platelet function in the presence of platelet-disrupting venom components botrocetin and convulxin, while aspercetin disrupts platelet and fibrin function similarly in opossum and human samples. Whole blood aggregometry and serum protease inhibition

**Data availability statement:** All raw data generated in this study, including microfluidic assay results, whole-blood aggregation data, and gelatinase inhibition assay outputs, including scripts for analysis and visualization are publicly available via the Zenodo Digital Repository: 10.5281/zenodo.15285263.

**Funding:** This work was supported by the NIH-NIGMS Minnesota IRACDA Program 2K12GM119955 (to D.H.D.); the NSF STEM-APWD DEB 2316784 (to D.H.D.), NIH R01-HL121212 (to A.S.G.), NIH R35-HL171421 (to D.G.), and NIH R35-HL150784 (to J.A.S.). The funders had no role in study design, data collection and analysis, decision to publish, or preparation of the manuscript.

**Competing interests:** The authors have declared that no competing interests exist.

showed patterns consistent with species-specific adaptation of mammals to their local snake venom. Unexpectedly, we find that *M. domestica* VWF requires increased shear force to elongate, a previously unknown aspect of opossum blood physiology that may contribute to venom resistance and may have relevance to human coagulopathies. Our findings demonstrate resistance under natural shear stress, and document venom resistance beyond large-bodied Didelphini, suggesting it is a widespread trait in South American marsupials.

## Introduction

Coevolution between enemies can drive dramatic molecular innovations, producing traits that depart radically from the physiological norms defined in model species like mice and humans [1,2]. When these evolved traits overlap with human-relevant pathways—such as hemostasis—they can yield not only evolutionary insight but unanticipated translational value [3]. Venomous animals and their prey offer some of the clearest examples of coevolutionary elaboration of traits, where reciprocal adaptations emerge even at deeply conserved molecular targets [reviewed in 4].

Venomous snakes and their mammalian predators are natural laboratories for studying coevolution [5,6]. In South America, members of the opossum clade Didelphini are well-known to hunt and consume pitvipers, including those from the genus *Bothrops*—the continent's most abundant and widespread group of venomous snakes [7]. Large-bodied opossums in this group exhibit remarkable venom resistance: they survive bites that would be fatal to other mammals, show minimal coagulopathic symptoms, and have evolved molecular defenses targeting venom components [8–10].

A key component of this resistance involves von Willebrand Factor (VWF), a critical blood-clotting protein that normally mediates platelet adhesion at sites of vascular injury—but is directly targeted by *Bothrops* venom to induce coagulopathy [11,12]. While many snake venom C-type lectin-like proteins (SNACLECs) with varying function are known, only botrocetin, aspercetin (from *Bothrops sp.*) and bitiscetin (from African *Bitis arietans*) have been identified to specifically target VWF [reviewed in 13]. Vessel damage will induce VWF to elongate and bind platelets in response to either exposed collagen or under increased shear force (caused by increased blood flow after vessel damage), creating a primary platelet plug preceding the formation of a more stable cross-linked fibrin clot [14,15]. Botrocetin acts as a coagulopathic intermediary, binding to VWF's A1 domain and platelet integrin GP1b to trigger platelet aggregation [11,12,16]. The resulting VWF–SNACLEC–platelet complexes are rapidly cleared from circulation, depleting both VWF and platelets and leading to hemorrhage [17]. This mechanism is further exacerbated by snake venom metalloproteinases (SVMPs), that cleave plasma proteins and break down vascular integrity to release other venom components like SNACLECs into circulation, further compounding coagulopathic damage [17,18].

In venom-resistant large-bodied opossums, VWF binds poorly *in vitro*—or not at all—to venom-derived SNACLECs like botrocetin, suggesting that specific mutations on opossum VWF A1 at the botrocetin binding site disrupt this toxin–target interaction

and contribute directly to resistance [10]. Surprisingly, VWF from several small-bodied opossum species also fails to bind botrocetin in protein-binding assays [10], and limited functional data from one small-bodied species (*Monodelphis domestica*) shows reduced platelet aggregation in static assays [9] despite no prior ecological or physiological evidence of venom resistance. These findings suggest that small-bodied species share venom SNACLEC resistance, but it remains unclear whether this reflects integrated, physiological resistance to snake venom that includes resistance to multiple protein classes.

While adaptive mutations under positive selection may point to coevolved traits, resistance at a single molecular interface does not establish a complex phenotype. Reductionist and static assays fail to capture the dynamic, integrated nature of coagulation in two important ways. Blood coagulation is a complex and dynamic process in which platelets are recruited by VWF to form a platelet plug that is stabilized by cross-linked fibrin [19]; therefore, the effects of introducing an antagonist to the system, such as botrocetin, should be evaluated for its immediate and downstream impacts. Notably, botrocetin can induce coagulopathy not just through canonical A1 domain binding, but via secondary mechanisms, including conformational changes in VWF and direct interactions with platelet receptors GP1b and $\alpha_{IIb}\beta_3$ that block the interaction of platelets and fibrinogen, disrupting hemostasis beyond directly bridging VWF and platelets [20].

Moreover, blood coagulation must function under the dynamic flow conditions of the vasculature that can alter the biochemical and biophysical properties of the system compared to closed "test-tube" experiments [21,22]. The flow-induced shear force—the force per unit area exerted by blood flow along vessel walls—is essential for activating VWF, yet has not been addressed in prior studies of opossum venom resistance. Shear forces initiate the unfolding of VWF multimers, exposing binding sites necessary for platelet capture and clot formation [23]. Opossum VWF that appears resistant under static conditions may become susceptible under shear, confounding inference of physiologic resistance. As such, opossums with VWF that fails to bind botrocetin under static conditions could still exhibit impaired platelet adhesion and fibrin formation under flow—functionally mimicking the response of susceptible species. Evaluating the response of VWF to venom SNACLECs under relevant vascular flow conditions is essential to understanding the extent to which coevolved resistance can maintain normal function in the presence of venom proteins.

Understanding the mechanistic basis of an extreme adaptation like venom resistance requires a systems-level approach, integrating multiple physiological processes to capture the full scope of how these animals survive repeated venom exposures. The current study examines physiological venom resistance to multiple coagulotoxic venom components in *M. domestica* under relevant flow states. We first examine the effects of the SNACLECs botrocetin, aspercetin, and convulxin on both platelet and fibrin deposition under physiologic shear and flow, and complement these assays with whole-blood platelet aggregometry analyses using these purified components as well as whole venoms from sympatric and allopatric vipers to test for signals of adaptation. We extend these tests of species specificity through assays of serum inhibition of the SVMPs to investigate the potential for adaptive, integrated resistance to multiple venom protein classes. We predicted that *M. domestica* would resist SNACLEC-induced disturbances under natural shear stress. Additionally, we expected elevated resistance to botrocetin and its source, *Bothrops jararaca* venom, including minimal disruption of platelet function and superior inhibition of SVMPs relative to humans and other mammals. We detect species-specific venom resistance across multiple venom protein types that aligns with the expected intensity of ecological interactions between snakes and mammals. Our data unexpectedly reveal altered *M. domestica* VWF elongation behavior under shear forces compared to human VWF, demonstrating how basic biological research on extreme adaptations can generate new knowledge of protein function and associated bleeding disorders.

## Materials and methods

### Whole venom and SNACLEC sourcing

Purified fractions of botrocetin from *B. jararaca* and aspercetin from *Bothrops asper,* bitiscetin from *Bitis arietans*, as well as whole venom from *B. jararaca* were used from previous work [9] (botrocetin B). Convulxin was purchased from ChemCruz (Santa Cruz Biotechnology Inc, Santa Cruz, CA). Whole venom from *Crotalus oreganus* and *Sistrurus miliarius* was

obtained during previous studies [24,25]. Venom SNACLECs were chosen to best provide a direct comparison of physiological and protein assays done in previous work [9,10].

## Opossum handling and blood collection

Opossums (n = 16) were housed in the Oberlin College research colony and cared for according to procedures previously described [26]. To collect blood for VWF purification, gray short-tailed opossums (*M. domestica*) were sacrificed by first being placed under isoflurane sedation and then exsanguinated via the posterior vena cava, per IACUC protocol F23BBYC-1 [27]. Opossum whole blood was collected in an acid citrate dextrose-coated syringe (ACD, Millipore Sigma) and stored in 1X ACD until use. Whole blood used for aggregometry assays were collected in 3.2% sodium citrate (Millipore Sigma). Human whole blood was collected from de-identified donors via venipuncture into 3.2% sodium citrate vacutainers (BD Pharmaceuticals) by the Case Western Reserve University Hematopoietic Biorepository and Cellular Therapy Core under an Institutional Review Board (IRB)-approved protocol (Case 12Z05, IRB # 09-90-195) of University Hospitals Cleveland Medical Center. Whole blood was fractionated by centrifugation at 150 rcf for 15 min at 25°C to isolate a platelet-rich plasma (PRP) supernatant, which was removed without disturbing the buffy coat. The PRP was further centrifuged at 2000 rcf for 25 minutes to recover platelet-poor plasma.

## VWF purification and analysis

Opossum VWF was purified by size-exclusion chromatography with Sepharose CL4B resin at 4°C as previously described [9,28] from plasma pooled after high-speed centrifugation. Protein containing fractions were monitored by absorbance at 280 nm using the in-line detector on the AKTA Pure. VWF containing fractions were pooled in dialysis tubing with a 10 kDa molecular weight cut of and concentrated by placing the tubing in Aquacide II (Millipore Sigma). VWF concentration was estimated by a custom alphaLISA [29] with the assumption that the VWF concentration in *M. domestica* is similar to that of humans. To compare opossums' VWF multimer structure to human, we performed a multimer analysis using VWF purified from *M. domestica* plasma and HumateP (a purified human plasma-derived product that contains VWF and Factor VIII, CSL Behring). Due to the large size of intact VWF multimers, agarose is used instead of polyacrylamide. VWF multimers were analyzed by electrophoresis using a 1.6% separating LDS agarose gel and visualized using the rabbit anti-human VWF antibody A0082 (DAKO) as previously described [30]. Protein was transferred to nitrocellulose using the semi-dry Power Blotter (ThermoFisher) and western blot was performed using the iBind Flex (Invitrogen) system with the rabbit anti-human VWF antibody A0082 (DAKO) as primary and IRDye 800CW anti-rabbit antibody (Li-Cor). Lanes were visualized a Li-Cor Odyssey CLx Imager and Image Studio (LiCor).

To ascertain that VWF from *M. domestica* bound to microchannel equine-derived type 1 fibrillar collagen (Chrono-log Corporation), purified and quantified opossum VWF was tagged with Alexa488 using the Alexa Fluor 488 (AF488) Microscale Protein Labelling Kit (Invitrogen) with minor modification for subsequent microchannel runs. The remaining untagged VWF was frozen and stored at -80C.

## Microfluidic assays

We performed microfluidic assays developed to approximate *in vivo* conditions and which allowed for real-time measurement of VWF/VWF-platelet recruitment and fibrin generation. Microfluidics experiments were performed in the Bioflux 200 system (Cell Microsystems) using 48-well High Shear Plates (Cell Microsystems) [31,32]. Glass-bottomed microfluidic channels with a width of 250 μm and a height of 70 μm were coated with 100 μg/mL type 1 equine fibrillar collagen (Chrono-log Corporation) in 20 μM acetic acid for 60 min at 37°C. These conditions facilitate the self-polymerization of collagen fibers which anchor to the glass substrate, creating an environment that mimics collagen exposure following vascular injury. Unbound collagen was washed by perfusing 0.1 wt% bovine serum albumin (BSA) in calcium-free phosphate buffered saline (PBS) at 30 dyn/cm$^2$ for 2 min.

While we have previously standardized this assay for use with human blood, it was unknown whether the equine collagen would allow binding of opossum VWF in a similar manner and under similar shear flow conditions. To verify that this assay is applicable to opossum VWF, we used purified and fluorescently tagged opossum VWF to determine experimental conditions and verify that opossum VWF binds effectively to equine fibrillar collagen used in standard microfluidics assays (Fig 1). AF488-VWF was perfused through the collagen-coated microfluidic channel at shear stresses ranging from 20 to 200 dyn/cm$^2$. VWF binding to collagen was imaged under epifluorescence using a Zeiss Axio Observer microscope with a 10x objective (Plan-NEOFLUAR, NA = 0.3). We observed similar overall VWF deposition, indicating that opossum VWF binds to equine collagen to a similar extent as human VWF.

With assay conditions established, we assessed the ability of *M. domestica* to resist the detrimental effects of venom SNACLECs. Opossum or human (as a venom non-resistant control species) whole blood was processed and used to perform assays within four hours of any given draw, to ensure that platelet activity in the blood remained viable. Citrated PRP was obtained as described in the *Opossum Handling and Blood Collection* section. PRP was incubated with 1 μg/mL calcein-AM (Invitrogen) to stain platelets and 10 μg/mL AF-647 fibrinogen (Invitrogen) for 30 min at room temperature. Samples were recalcified to 20 mM CaCl$_2$ and immediately perfused through collagen-coated channels at 60 dyn/cm$^2$, a shear condition that is relevant to wound environments. These conditions provide more fine-tuned information about the degree and type of physiological response induced by exposed collagen in the presence of agonists. Epifluorescence microscopy was used to image platelet accumulation and fibrin generation in real time. Samples were run without any additives (Control) to assess normal platelet adhesion and fibrin generation. For each sample, complementary assays were performed in the presence of three venom derived SNACLECs known to affect platelet aggregation function: botrocetin, aspercetin, and convulxin. Each SNACLEC was used at its ED$_{90}$ concentration for human blood: 4 μg/mL for botrocetin, 20 μg/mL for aspercetin, and 0.5 μg/mL for convulxin. In this assay, we expect that SNACLEC-induced VWF/platelet complexation leading to thrombocytopenia and low VWF levels should result in reduced platelet adhesion and prolonged fibrin generation times, mimicking bleeding observed *in vivo*. To account for differences in platelet counts between individual animals, platelet adhesion was normalized to the control condition for each sample.

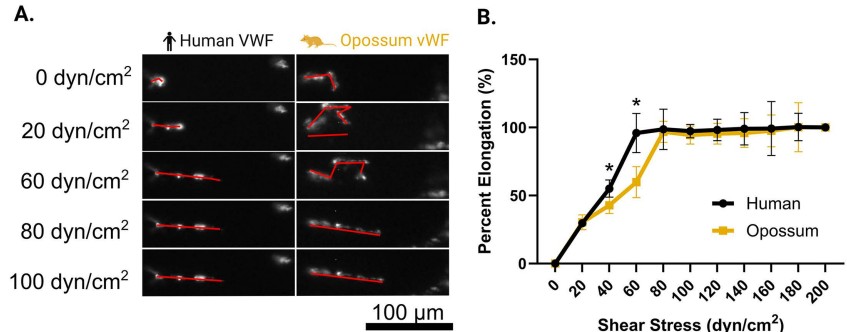

**Fig 1. Shear-dependent elongation of von Willebrand Factor (VWF) in human and *Monodelphis domestica*.** (A) Elongation of von Willebrand Factor (VWF) under different shear conditions. (B) The degree of elongation for both human and *Monodelphis domestica* (opossum) VWF under varying shear forces in a microfluidics assay. Purified VWF was fluorescently tagged and perfused across collagen-coated channels at shear rates of 0 dyn/cm$^2$ to 200 dyn/cm$^2$ for human and opossum samples, respectively. The percentage elongation of VWF was quantified and normalized to both the 0 dyn/cm$^2$ condition (0% elongation) and the 200 dyn/cm$^2$ condition (100% elongation). At 60 dyn/cm$^2$, human VWF showed full elongation, whereas opossum VWF remained partially coiled, indicating that higher shear rates are required to induce full elongation in the latter. When perfused at 80 dyn/cm$^2$, opossum VWF fully elongated, mirroring the response observed in human VWF at 60 dyn/cm$^2$. Error bars represent the average ± SD of n = 10 replicates. Asterisks denote statistically significant differences in percent elongation (p-value <0.05). High degrees of fluorescent labeling caused aggregation or "clumping" of VWF multimers in both human and possum samples, inhibiting complete elongation. Reduced labeling allowed full elongation for human VWF at 60 dyn/cm$^2$, but opossum VWF still required the higher shear rate (80 dyn/cm$^2$) to achieve comparable uncoiling.

## Aggregation assays

To test *M. domestica*'s ability to resist whole venom, as well as isolated SNACLECs, whole blood aggregation was assessed using an impedance aggregometer (Roche multiplate aggregometer). Briefly, these assays differ from microfluidic assays as coagulopathic response is indicated by *induction* of platelet aggregation, rather than inhibition. In the absence of exposed collagen and/or shear force, platelets in blood (in a stationary cuvette) should not rapidly aggregate unless exposed to a platelet activating agonist (i.e., whole venom or pure venom-derived SNACLECs). Failure of agonists to induce platelet aggregation suggests strong venom resistance to coagulopathic venom components, including SNACLECs. Thus, we expect *M. domestica*'s platelet aggregation response to be lower for sympatric venom and SNACLECs than for allopatric venom and SNACLECs, suggesting higher resistance to species they have evolved with.

Platelet aggregation response in this assay is measured by impedance, where higher impedance means a greater response. Purified SNACLECs used were aspercetin (20 μL, 0.1 mg/mL), convulxin (20 μg, 0.01 mg/mL), and botrocetin (20 μg, 0.01 mg/mL). Platelet aggregation was measured over 10 minutes, with impedance changes recorded as a measure of platelet aggregation. All purified SNACLECs were selected from South American vipers that were either completely sympatric (*B. jararaca*), partially sympatric (*Crotalus durissus*), or allopatric (*Bothrops asper*) with *M. domestica*. It should be noted that while aspercetin and botrocetin induce aggregation via VWF, convulxin induces aggregation via a direct interaction with the platelet. In addition to purified SNACLECs, we also challenged *M. domestica* with whole venom from the same sympatric species (*B. jararaca*), as well as two distantly related allopatric species with vastly diverse venoms (*C. adamanteus* and *S. miliarius*) to provide some measure of the specificity/cross-reactivity of venom resistance in *M. domestica* in the absence of availability of whole venom from *B. asper* and *C. durissus* (see Fig 3 range maps).

Venous blood samples collected from *M. domestica* were stored at room temperature in 3.2% sodium citrate and tested within four hours of collection. For each assay, 300 μl citrated whole blood and 300 μl NaCl/CaCl$_2$ solution (0.9% saline and 3mM CaCl$_2$ provided by the manufacturer) were aliquoted into cuvettes and stirred at 37°C. Aggregation was induced by adding 20 μL of agonists to the samples. Collagen (20 μg/mL) was used as a positive control, PBS (phosphate buffered saline) was used as a negative control. Aggregation response was calculated as the average of two replicate measures. To facilitate cross-comparison, the data were normalized by subtracting the minimum value within each treatment group, ensuring that the minimum impedance value was zero. Data visualization was performed using R software to allow for comparison of platelet responses to different agonists. Venom assay and controls were conducted in triplicate to ensure reproducibility; however, due to limited plasma volume, SNACLEC tests were not replicated.

## Venom SVMP inhibition assays

We evaluated the inhibition of SVMPs in whole venom by *M. domestica* serum against three snake venoms (*B. jararaca*, *C. oreganus*, and *B. arietans*) using a modified protocol from the EnzChek Gelatinase/Collagenase Assay Kit (Life Technologies, Carlsbad, CA, USA). Briefly, this assay tests the ability of *Monodelphis domestica* serum to neutralize venom-induced degradation of a gelatin substrate, which mimics the extracellular matrix proteins targeted by SVMPs [33]. The DQ-gelatin substrate was diluted to 1:20 in reaction buffer for the assay.

Serum was taken from opossums during venous blood extraction and frozen for subsequent analyses, and the protein concentration of each serum sample determined with the Pierce BCA Protein Assay Kit. Venom samples were diluted to a final protein concentration of 0.3125 ng per well. Venom was incubated with various amounts of *M. domestica* serum, with 0, 5, 10, 25, 50, 125, 250, or 475 μg serum protein per well, to determine the extent of venom inhibition. Human serum was used as a non-coevolved venom-sensitive baseline for venom SVMP inhibition by serum proteins, and California ground squirrel (*Otospermophilus beecheyi*) was used to compare species-specific inhibition as it has previously been shown to have resistance for its sympatric predator (*C. oreganus*) [25].

Fluorescence intensity was measured in relative fluorescence units (RFU) using a SpectraMax M2 microplate reader (Molecular Devices) at 30 second intervals. The slope of the fluorescence increase (RFU min$^{-1}$) from the linear portion of the reaction curve was calculated to quantify the venom's SVMP activity. Decreased slope values, as compared to control reactions with no serum, indicate successful inhibition of venom enzymatic activity by the serum. Each experiment was performed in triplicate, and data were normalized to the control (untreated venom) to calculate relative SVMP activity..

### Statistical analyses

For elongation assays under varying shear stress conditions, percentage elongation was normalized relative to the 0 dyn/cm² condition (set as 0% elongation) and the 200 dyn/cm² condition (set as 100% elongation) for each species. Data are presented as mean ± standard deviation (SD) based on $n = 10$ replicates per condition. Significance was assessed using Welch's unpaired t-test.

For comparisons of platelet adhesion and fibrin lag time, statistical significance was assessed using one-way ANOVA with Fisher's Least Significant Difference (LSD) post-hoc correction. Significance thresholds were defined as follows: $p \leq 0.05$ (*), $p \leq 0.01$ (**), $p \leq 0.001$ (***), and $p \leq 0.0001$ (****). Data are presented as mean ± standard deviation (SD). Platelet adhesion values were normalized to the control condition for each individual sample to account for variation in platelet counts across donors or animals. Fibrin fluorescence intensity was normalized to background fluorescence (0) and maximum fluorescence (1).

For SVMP inhibition assays, dose-response curves were fit using the drc package [34] in R, applying a four-parameter logistic model to serum treatments from *Monodelphis domestica*, *O. beecheyi*, and *H. Sapiens* sera. $ED_{90}$ values were calculated from these fitted curves to quantify the serum volume required to inhibit 90% of venom gelatinase activity.

## Results

### Comparative microfluidic analysis of VWF behavior

To determine whether evolved sites in *M. domestica* VWF [as described in 10] result in functional resistance under physiologically relevant conditions, we first assessed VWF elongation, platelet adhesion, and fibrin formation using microfluidic assays that simulate *in vivo* shear forces. In humans, higher molecular weight VWF multimers are more procoagulant than lower molecular weight forms as they are more likely to bind platelets after interacting with subendothelial collagen under flow. Before proceeding, we confirmed the existence of high molecular weight multimers in purified VWF from *M. domestica* and compared it to that of a human-derived VWF product that mimics the human plasma distribution. We confirmed the existence of high, intermediate, and low molecular weight multimers in our *M. domestica* VWF preparation (S1 Fig). Under shear, we observed a surprising difference in the dynamics of elongation between *M. domestica* and human VWF (Fig 1). Compared to human VWF, opossum VWF required an additional 20 dyn/cm² to fully elongate, and demonstrated statistically reduced elongation at shear stresses of 40 and 60 dyn/cm².

### Microfluidic comparison of venom effects on platelet adhesion and fibrin generation

We next compared the platelet adhesion response of opossum and human PRP to purified venom SNACLECs to evaluate whether resistance is preserved in integrated clotting dynamics under flow. Control experiments with human PRP showed that platelet accumulation on the microfluidic surface was strongly inhibited by botrocetin and convulxin, and weakly inhibited by aspercetin (Fig 2A–D), consistent with prior work demonstrating that botrocetin and convulxin strongly activate or disrupt platelet function in human blood [35,36]. Venom derived SNACLECs also significantly reduced platelet fluorescence in *M. domestica* PRP (Fig 2E–H); however, the defect was less pronounced after treatment with botrocetin or convulxin than in human PRP (Fig 2B,F,D,H), suggesting opossum resistance to those SNACLECs. In human PRP, the initial rate of platelet adhesion was similar for the control, botrocetin and aspercetin conditions, but reduced to almost zero in

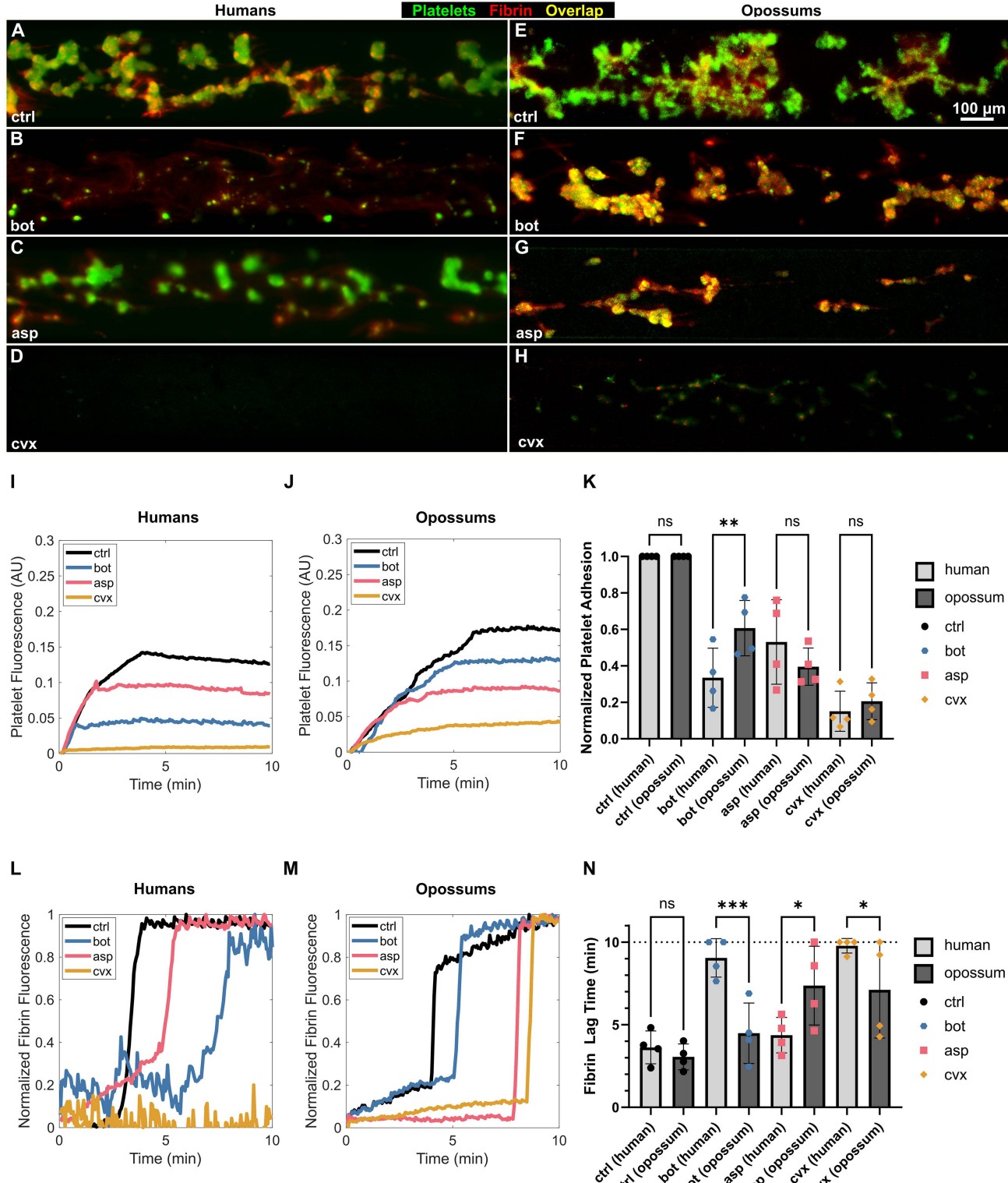

**Fig 2. *Monodelphis domestica* platelets are resistant to botrocetin but not aspercetin or convulxin.** Compared to humans, opossums exhibited resistance to botrocetin (bot), but not aspercetin (asp) or convulxin (cvx). A-H) Endpoint images of platelet (fluorescently labeled in green) adhesion to a collagen coated substrate and fibrin (fluorescently labeled in red) generation after 10 min of shear flow at 60 dyn/cm². A) Human platelet rich plasma (h-PFP) perfused across a collagen-coated substrate resulted in robust platelet adhesion in the absence of venom derivatives (ctrl). B) Platelet adhesion

was significantly reduced in the presence of bot, C) moderately reduced in the presence of asp, D) and almost completely abrogated in the presence of cvx. E) Compared to h-PRP, opossum PRP (o-PRP) exhibited a similar extent of platelet adhesion to collagen. F) However, platelet adhesion was affected to a lesser extent by bot. H) o-PRP responded similar to h-PRP to asp, but H) platelet adhesion in o-PRP was not entirely abolished by cvx. I-J) Representative quantification of platelet fluorescence over time. I) In h-PRP, the overall extent of platelet adhesion was reduced slightly by asp, significantly by bot, and almost completely by cvx. J) Though platelet adhesion in o-PRP was affected by asp to a similar degree as h-PRP, platelet adhesion was only barely reduced by bot. Some degree of platelet adhesion was preserved in the presence of cvx. K) Aggregate platelet adhesion data, normalized to the ctrl condition for each individual donor or opossum. The defect in platelet adhesion was similar for h-PRP and o-PRP for asp and cvx, but reduced significantly in o-PRP for bot. L-M) Representative quantification of fibrin(ogen) fluorescence over time. The sharp ramp indicates fibrin generation, and the corresponding time point is parameterized as fibrin lag time. L) In h-PRP, fibrin polymerized quickly in the absence of venom derivatives, owing to the procoagulant surface of collagen-bound platelets. Asp and bot both delayed the formation of fibrin, with bot having a more pronounced effect. Fibrin did not polymerize during the 10 min experiment in the presence of cvx. M) In o-PRP, bot caused only a marginal prolongation in fibrin lag time. Both asp and cvx significantly delayed fibrin generation. N) Aggregate fibrin generation data. The dashed line at t = 10 min indicates the endpoint of the experiment. On average, fibrin generated in less than 4 min in h-PRP. Bot more than doubled fibrin lag time, with half of the replicates not generating fibrin within the 10 min experimental window. Despite its similarity to bot, asp did not significantly affect fibrin lag time. In three of four replicates, fibrin did not generate within 10 min in the presence of cvx. In o-PRP, fibrin lag time was prolonged by asp and cvx, but not bot. o-PRP generated fibrin faster than h-PRP in the presence of bot or cvx, but slower than h-PRP in the presence of asp. Statistics were performed using one-way ANOVA using Fisher's LSD correction. Error bars mean ± SD. For significance, *: $p \leq 0.05$, **: $p \leq 0.01$, *** $p \leq 0.001$, ****$p \leq 0.0001$.

the presence of convulxin (as indicated by the initial slope of the platelet fluorescence curves, Fig 2I). A similar trend was observed in opossum PRP, except that platelets adhered even in the presence of convulxin, though at much lower levels than the other conditions (Fig 2J). The effect of botrocetin on platelet adherence to the microchannel was decreased in *M. domestica* (Fig 2B,F,J): botrocetin caused a more severe defect in the total extent of platelet adhesion than aspercetin in human PRP, but a less severe defect than aspercetin in opossum PRP (Fig 2I–K). Aspercetin and convulxin caused a similar response in both species (Fig 2K). The effect of botrocetin on platelet adherence to the microchannel was also significantly decreased in *M. domestica* when compared to the same in human (Fig 2B,F,K). In human PRP, botrocetin caused a 66 ± 16% reduction in the overall extent of platelet adhesion compared to the negative control, whereas it only caused a 39 ± 15% reduction in opossum PRP (Fig 2K).

Due to the complex nature of blood coagulation, we quantified not only the effects of SNACLECs on platelet adhesion but also fibrin deposition so as to evaluate downstream coagulopathic disruptions. We specifically quantified fibrin generation kinetics (by quantifying 'lag time' reflecting the time needed for fibrin clot formation), where a prolonged lag time is unfavorable for effective hemostasis and predictive of bleeding. Purified venom SNACLECs increased lag time in both humans and opossums (Fig 2L–N). While convulxin had a dramatic effect in human samples, abrogating fibrin formation almost completely, it showed a more modest response in opossums (Fig 2L–N). As with platelet adhesion, the greatest signal for resistance was seen in botrocetin treatments (Fig 2L–N), with an overall reduced effect of botrocetin in opossum PRP (Fig 2L–N). Botrocetin caused a nearly threefold increase in fibrin lag time from 3.6 ± 1.0 min to 9.0 ± 1.2 min in human PRP, but did not significantly prolong fibrin lag time in opossum PRP (from 3.1 ± 0.8 to 4.5 ± 1.8 min, Fig 2N).

Aspercetin did not significantly increase the fibrin lag time for human PRP but doubled lag time from 3.1 ± 0.8 min to 7.4 ± 2.4 min in opossum PRP, which is perhaps suggestive of opossum susceptibility to aspercetin. This result is consistent with previously established weak affinity of aspercetin to VWF A1 in protein binding assays [37].

Convulxin induced a reduced coagulopathy in opossum PRP compared to human PRP but remained the most potent effector of integrated hemostatic function in both species. Platelet fluorescence and adhesion in response to aspercetin was roughly similar between human and opossum, though aspercetin had a slightly stronger effect on fibrin lag time in opossums. Compared to humans, *M. domestica* VWF showed a measurably reduced integrated coagulopathic response (platelet fluorescence, platelet adhesion, and fibrin lag time) to botrocetin, suggesting that these opossums have functionally meaningful hemostatic protective effects against this SNACLEC.

## Whole blood aggregometry

To further assess species-specific resistance to venom in the milieu of whole blood, we used whole blood aggregometry to test the platelet aggregation response of *M. domestica* to both whole venom and purified SNACLECs. We used impedance aggregometry to further probe the species-specificity of opossum resistance to disrupted platelet function. These tests revealed differential, species-specific platelet responses by *M. domestica* to the whole snake venoms and purified SNACLECs tested (Fig 3). As shown in Fig 3A, collagen, used as a positive control, induced a rapid and sustained

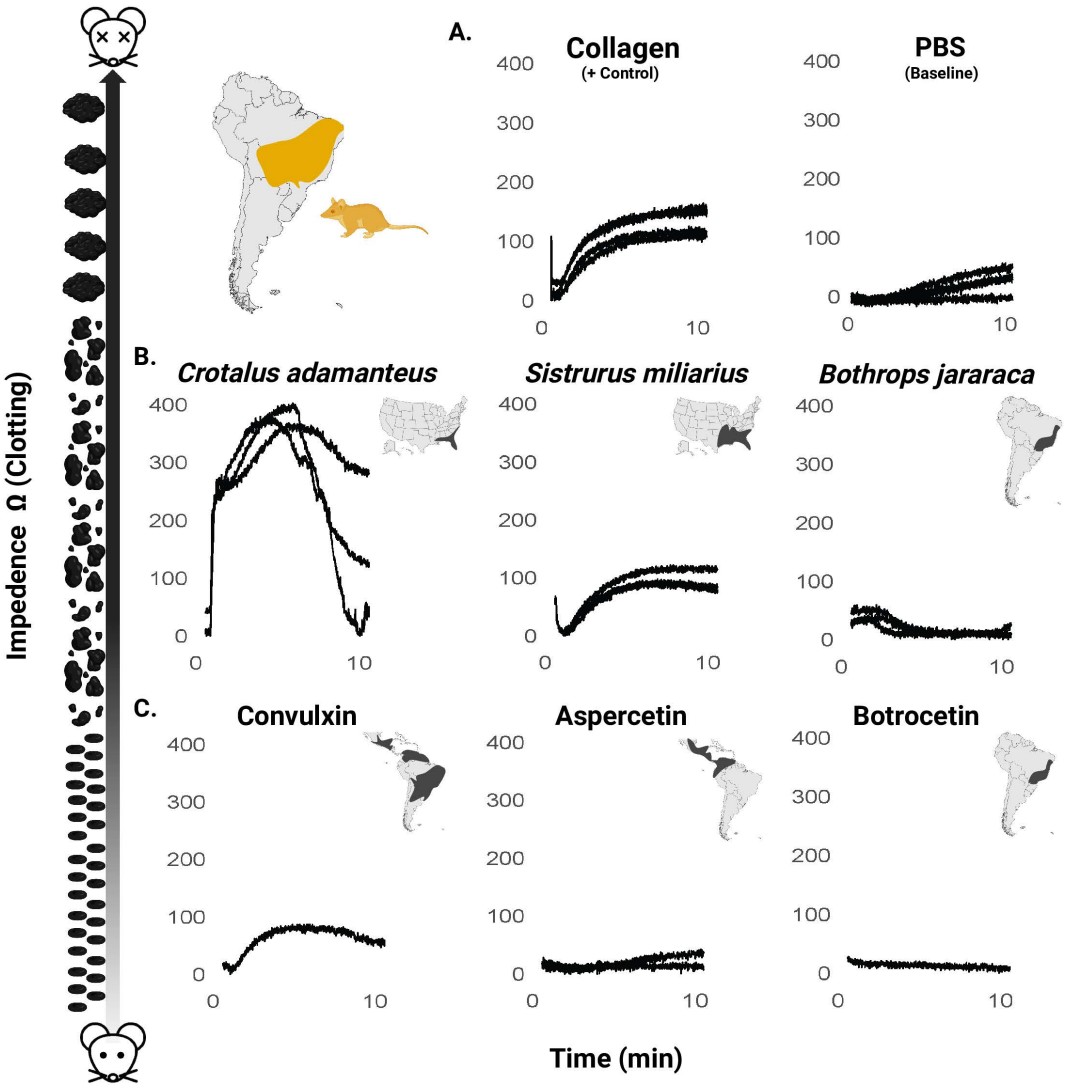

**Fig 3. Platelet aggregation *in Monodelphis domestica* whole blood in response to venoms and purified C-type lectin-like proteins.** Impedance was measured over a 10-minute period to assess platelet aggregation response. (A) Collagen (positive control), and PBS (negative control). (B) Whole venoms (*Crotalus adamanteus*, *Sistrurus miliarius*, and *Bothrops jararaca*) are in the middle panels. (C) C-type lectin-like proteins (SNACLECs), Convulxin (derived from *Crotalus durissus* venom), aspercetin (derived from *Bothrops asper* venom) and botrocetin (derived from *B. jararaca* venom) are on the bottom panels. Data represents the average impedance change over time, normalized to account for baseline variation. Multiple curves represent independent samples (replicates) performed when available. Cartoon on the left depicts platelet aggregation (lowest at y = 0 and increases at higher values of y). Range maps of species from which each venom/SNACLEC are derived are shown in the upper right of each graph to allow for a comparison of aggregation response with sympatry. Range maps were created with BioRender.com with data derived from https://www.iucnredlist.org/ [38–41].

increase in impedance, indicative of expected collagen-induced platelet aggregation, while PBS (negative control) established a baseline throughout the 10-minute measurement period.

Among whole venoms, allopatric *C. adamanteus* venom elicited the most pronounced platelet aggregation response, with impedance peaking around 5 minutes at approximately 220% of the collagen control (400 Ω) before showing pronounced disaggregation, indicating a robust but transient aggregation effect (Fig 3B). Allopatric *S. miliarius* venom induced a response reaching approximately 70% of the collagen control (180 Ω), while sympatric *B. jararaca* venom resulted in a minimal increase in impedance, remaining just slightly below the PBS baseline (50 Ω), suggesting substantial resistance to platelet activation (Fig 3B). As expected, collagen (positive control) produced a peak impedance of 180 Ω, while PBS (baseline control) showed negligible effects with impedance remaining below 50 Ω, which is consistent with standard values for other mammals [42,43].

The purified SNACLECs demonstrated variable effects on platelet aggregation. Convulxin, a known collagen-like agonist, induced a notable increase in impedance, mirroring the aggregation pattern of the collagen control but with a delayed onset (Fig 3C). Consistent with whole venom results which showed no resistance to *Crotalus* venom, this indicates a lack of resistance to Convulxin (derived from *Crotalus durissus* venom). Aspercetin, which targets VWF, triggered a measurable but extremely reduced platelet aggregation response. Botrocetin, like whole *B. jararaca* venom, elicited no detectable platelet aggregation response. Due to limited quantities of these purified SNACLECs, we prioritized their use in *Monodelphis domestica* whole blood. Human platelet responses to these same protein fractions have been previously characterized in PRP [9,16], and we refer readers to those studies for comparative context. The responses of both aspercetin and botrocetin observed here are extremely minimal compared to previously well-established responses for susceptible species, suggesting strong resistance. Overall, *M. domestica* showed the lowest levels of whole blood aggregation by both snake venom and purified SNACLECs from their sympatric snake predators in the genus *Bothrops* and appeared to be entirely resistant to both platelet disturbance by *B. jararaca* venom and botrocetin. This is consistent with the hypothesis of evolution of increased resistance to venom of sympatric snakes in *M. domestica*.

## Venom metalloproteinase inhibition

To evaluate whether resistance in *M. domestica* extends beyond VWF-targeting SNACLECs and to test for ecological specificity of serum-based SVMP inhibition, we conducted a complete reciprocal cross of 3 viperid species venoms with 3 mammal serums. Venoms were selected to represent 1) a sympatric species (*B. jararaca*) for which ecological and other data predict opossum resistance, 2) an allopatric species (*C. oreganus*) for which another mammal (ground squirrels) have evolved venom resistance 3) an allopatric distantly related viper (*B. arietans*) (see Fig 4 top and middle panels).

Blood serum from *M. domestica* demonstrated varying levels of inhibition across the three snake venoms tested: *B. jararaca*, *C. oreganus*, and *B. arietans* (Fig 4A–C; Table 1). In all cases, increasing concentrations of serum protein led to a dose-dependent decrease in venom proteinase activity, indicating the presence of SVMP inhibitors. For *B. jararaca* venom (Fig 4A), *M. domestica* serum rapidly inhibited SVMP activity at the middle to high concentrations of serum protein tested, resulting in the lowest $ED_{90}$ value of the three sera tested, indicating it inhibited venom approximately 2.5 times better than squirrel serum and 14 times better than human serum (Table 1). Similarly, serum of the squirrel *O. beecheyi* yielded a lower $ED_{90}$ for its sympatric predator, *C. oreganus*, than did the human or opossum sera (Table 1), suggesting reciprocal specificity of these two resistant mammals for their sympatric snake predators. Finally, *M. domestica* serum outperformed human and *O. beecheyi* serum against the venom of the African viper *B. arietans*. Enhanced inhibition for local/sympatric venoms combined with a more general pattern for the African viper venom supports coevolved specificity of mammal serum inhibitors and SVMPs.

                                                                    

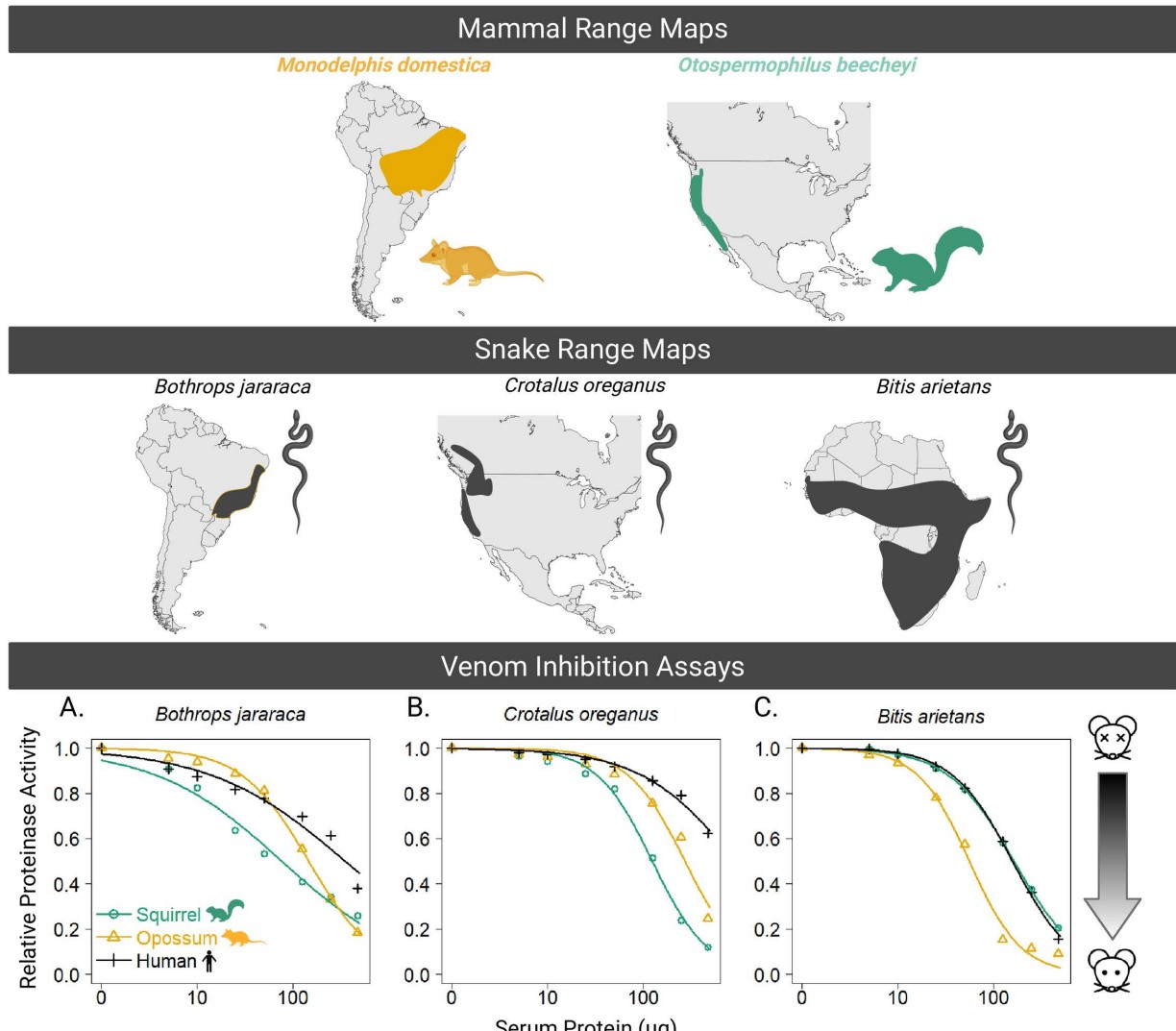

**Fig 4. Geographic distributions of mammals (*Monodelphis domestica*, *Otospermophilus beecheyi*) and snakes (*Bothrops jararaca*, *Crotalus oreganus*, *Bitis arietans*) alongside venom inhibition assay results.** Range maps for mammals *M. domestica* and *O. beecheyi* (top), and snakes (*B. jararaca*, *C. oreganus*, and *B. arietans* (middle). Bottom panel shows results of the venom inhibition assays. *Bothrops jararaca*, *Crotalus oreganus*, and *Bitis arietans* venoms were incubated with increasing concentrations of mammalian serum (*M. domestica*, *O. beecheyi*, and *H. sapiens*), and relative degradation of a gelatin substrate was measured. Data points represent the mean ± SEM for three independent experiments. *M. domestica* serum demonstrated greater inhibition of proteinase activity compared to squirrel and human sera, at higher serum concentrations, particularly for venoms from *B. jararaca* and *B. arietans*. Inhibition was assessed based on the reduction in fluorescence due to the degradation of a gelatin substrate, reflecting the venom's proteolytic activity. In this graph, the x axis would represent total resistance (depicted with the alive mouse icon), and decrease as you move away from y = 0 (depicted by the dead mouse icon). Range maps were created with BioRender.com with data derived from https://www.iucnredlist.org/ [40,41,44,45].

## Discussion

Understanding the physiological mechanisms, molecular underpinnings, and evolutionary history of venom resistance across opossums is crucial for understanding how selective pressures, ecological interactions, and physiological constraints shape the evolution of such extreme adaptations [46,47]. Our results provide compelling evidence that *M. domestica* possesses considerable resistance to both SNACLECs and whole venom from *B. jararaca*, and suggests that

**Table 1. Species-specific ED$_{90}$ values (µg/mL) derived from dose-response analysis of SVMP inhibition by increasing amounts of serum protein from three mammal species. ED$_{90}$ (Effective Dose, 90%) indicates the amount of serum required to inhibit 90% of the venom's proteinase activity. Asterisks indicate species that are sympatric with the snake venom tested.**

| Source of Venom | Serum | ED$_{90}$ | S.E. |
|---|---|---|---|
| *Bothrops jararaca* | Human | 12379.2 | 6272.4 |
| | *M. domestica*[*] | 865.4 | 149.1 |
| | *O. beecheyi* | 2182.2 | 636.7 |
| *Crotalus oreganus* | Human | 11391.6 | 7660.1 |
| | *M. domestica* | 1236.6 | 231.7 |
| | *O. beecheyi*[*] | 541.9 | 71.4 |
| *Bitis arietans* | Human | 774.4 | 80.1 |
| | *M. domestica* | 220.5 | 21.7 |
| | *O. beecheyi* | 940.5 | 107.7 |

previous *in vitro* work showing convergently derived mutations on VWF is indicative of realized venom resistance [37]. We also uncover a previously undescribed modification to opossum VWF: that it requires a greater shear force to elongate, potentially contributing to the resistance phenotype or other important physiologic variation.

Under natural flow and shear, we uncovered different responses of *M. domestica* and *H. sapiens* samples to the three venom SNACLEC proteins, with opossums strongly resisting botrocetin, mildly resisting convulxin, and showing no resistance to aspercetin. Aspercetin is a VWF-mediated agonist from a closely related viper (*Bothrops asper*) and is thought to work in a similar manner to botrocetin because of its ability to induce VWF-mediated platelet aggregation, but detailed structural studies have yet to locate its exact binding site on VWF and prior kinetics studies show weak binding to the A1 domain [16,37]. Such differential susceptibility to related toxins is the expectation under coevolutionary dynamics such as molecular matching-based arms-races or "Red Queen" coevolution, where aspercetin may represent coevolutionary "escape" from opossum resistance mechanisms. Alternatively, the comparison between botrocetin and aspercetin may represent adaptation to the botrocetin of sympatric *B. jararaca*, while a lack of interaction with allopatric *B. asper* leads to a lack of resistance to aspercetin.

Consistent with previous work on botrocetin resistance in Didelphini, *M. domestica* showed significantly reduced perturbation of both platelet accumulation and fibrin deposition under flow, suggesting that *M. domestica* maintains hemostatic competence when challenged by this toxin from *B. jararaca*. Botrocetin is well-known for inducing coagulopathy by binding to VWF (at the A1 domain) and inducing VWF to elongate and attach to platelets [12,11]. Botrocetin then changes conformation slightly and binds secondarily to the platelet GP1b binding site, creating a tight trimolecular complex which renders platelets unavailable for adhesion and causes extreme thrombocytopenia [11,48]. However, botrocetin was recently revealed to act independently of VWF via platelet integrin αIIbβ3 to block binding of fibrinogen and other ligands essential for coagulation [20]. More detailed work will be needed to understand whether resistance includes both the VWF-dependent and independent modes of botrocetin action.

To our knowledge, the potential for convulxin resistance in wild mammals has not been previously studied. Convulxin is derived from the more distantly related neotropical rattlesnake *C. durissus*, though there are concrete reports of predation on *C. durissus* by opossums [49] and thus the opportunity for natural selection. Convulxin binds directly to the GPVI platelet receptor, inducing coagulopathy by a mechanism independent of VWF [50]. The lack of a difference in performance concerning *M. domestica* and *H. sapiens* platelet adhesion paired with a modest difference between these mammals in the convulxin-induced fibrin lag time may be consistent with evolved resistance to impacts on fibrin formation in opossums, though we note high variability among *M. domestica* replicates and that fibrin lag time still doubled relative to the control condition. Taken together, the evidence here points to susceptibility of *M. domestica* to the effects of convulxin.

Our assays of whole blood impedance aggregation revealed a reduced response of *M. domestica* blood to both botrocetin and aspercetin compared to both the collagen control and convulxin. Initially, this might lead us to assume that opossum resistance is specific to *Bothrops* SNACLECs. However, microfluidic assays showed that platelet adhesion and fibrin clot formation in opossums were more disrupted by aspercetin than botrocetin. Aspercetin appears to be able to function nearly as well in opossums as it does in humans under flow (in both cases it has weak function), but not in whole blood or previous PRP assays [10], highlighting the importance of integrated assays that incorporate the complex milieu of the blood coagulation system as well the dynamic shear forces that act upon [9,16,37]. Aspercetin may act directly on platelets, fibrin, or other agonists more strongly during platelet adhesion and fibrin clot formation, or its function in opossums may depend on the presence of shear force. Another unexplored possibility as that aspercetin's affinity for VWF is attuned to a VWF target from another species of predator or prey.

We found evidence in both whole blood aggregometry and SVMP inhibition assays for adaptive species specificity of *M. domestica* in resisting whole snake venoms. First, we observed a decreased aggregation response in *M. domestica* blood (to whole venoms from *Bothrops jararaca*, but not allopatric *Crotalus adamanteus* or *S. miliarius* venoms). Second, *M. domestica* serum showed superior ability to inhibit the SVMP enzymatic action of *B.jararaca* venom, while *C. oreganus* was best inhibited by its sympatric, venom resistant prey (*O. beecheyi*). This serum inhibition test is powerful because it employed a fully reciprocal cross of the venoms and sera. The pattern of adaptive specificity adds to similar work using the venoms of two rattlesnake species and their primary squirrel prey in suggesting that, for species-level divergence and allopatric distributions, prey resistance is adapted to inhibiting the local snake venom SVMPs [51]. *M. domestica* serum also most strongly inhibited venom from the African viper, *B. arietans*, suggesting that *M. domestica* also possesses generalist SVMP inhibitor proteins, similar to those seen in other highly resistant taxa [7,52].

Finally, we found that VWF in *M. domestica* requires a comparatively higher shear force than that for human VWF, to elongate. We hypothesize that this property could be a correlate of venom resistance by making it more difficult for venom components, like botrocetin, to bind VWF and exact subsequent coagulopathic effects. Elevated shear thresholds for VWF elongation may also protect against other physiological disruptions, such as spontaneous platelet adhesion in high-flow conditions or inappropriate clot formation. Beyond venom resistance, this trait may be linked to broader physiologies of non-placental mammals, potentially reflecting differences in vascular dynamics, metabolism, or reproductive strategies. The discovery of this and potentially other species with shear-resistant VWF may also provide a novel avenue for therapeutic exploration, where inducing similar shear-resistance in VWF may prevent bleeding disorders associated with left ventricular assist devices (LVAD) [53]. While there is some limited data to suggest that increased shear elongation may occur in other venom resistant species (pigs), little is known about the diversity of this trait across species, or what potential costs and tradeoffs it might involve [54].

Our work adds detailed comparative assays using physiologic vessel models to demonstrate that venom resistance is present in small-bodied opossums, and highlights the ability of comparative analyses to inform otherwise cryptic adaptations and ecologies [6,9,37]. Opossum species with similar and convergently evolved changes in VWF may have similar degrees of venom resistance [37], meriting further work integrating molecular, ecological, and physiological data in these understudied mammals. Our results also provide vital insight into the evolution of venom by suggesting that South American viper venoms may have evolved in the presence of many more species of co-evolving venom-resistant mammals than previously assumed. Future work on the molecular basis and evolution of venom resistance should recognize the importance of leveraging new technologies that can replicate *in vivo* physiological conditions while facilitating controlled, multi-species comparisons.

## Supporting information

**S1 Fig.** Opossum VWF Multimer Gel 1.6% resolving + 0.8% stacking Tris Glycine LDS-SeaKem HGT Agarose Vertical Multimer Gel (8x8cm vertical). Multimeric structures of von Willebrand factor (VWF) were analyzed using a 1.6% lithium dodecyl sulfate (LDS) agarose gel under non-reducing conditions. Lane 2 contained 30 µL of purified Monodelphis

domestica VWF (heated at 60°C for 30 minutes) and 10µl loading buffer. Lane 5 contained 30 µL HumateP (human VWF) as a control (heated at 60°C for 30 minutes) and 10µl loading buffer. The gel was transferred to a nitrocellulose membrane and visualized by fluorescence with anti-human VWF antibody (Dako A0082) using a Li-Cor Odyssey CLx Imaging System (Li-Cor).
(TIF)

**S1 Raw Images.** Original, unaltered image corresponding to the VWF multimer gel shown in S1 Fig. Includes full gel context and supporting image data used for visualization.
(TIF)

## Acknowledgments

The authors would like to thank Alexandra Rucavado, Erika Hingst-Zaher, and the Kentucky Reptile Zoo for their help in acquisition and purification of venoms used for this work. The authors would like to thank Suzanne McGaugh for tremendous guidance as well as lab and resource support. The authors would also like to thank Marvin Neiman (Case Western), Alison Narayan/Sarah Ackenhusen (University of Michigan), and Randy Westrick (Oakland University) for their guidance, feedback, and support.

## Author contributions

**Conceptualization:** Matthew L. Holding, Dante Disharoon, Laura M. Haynes, Anirban Sen Gupta, Yolanda Cruz, Danielle H. Drabeck.

**Data curation:** Danielle H. Drabeck.

**Formal analysis:** Matthew L. Holding, Dante Disharoon, Laura M. Haynes, Bipin Chakravarthy Paruchuri, M. Hao Hao Pontius, Krista Golden, Karl C. Desch, Yolanda Cruz, Danielle H. Drabeck.

**Funding acquisition:** David Ginsburg, Anirban Sen Gupta.

**Investigation:** Matthew L. Holding, Dante Disharoon, Laura M. Haynes, Bipin Chakravarthy Paruchuri, M. Hao Hao Pontius, Karl C. Desch, Anirban Sen Gupta, Yolanda Cruz, Danielle H. Drabeck.

**Methodology:** Laura M. Haynes, Bipin Chakravarthy Paruchuri, Krista Golden, Danielle H. Drabeck.

**Project administration:** Matthew L. Holding, Laura M. Haynes, David Ginsburg, Danielle H. Drabeck.

**Resources:** Jordan A. Shavit, David Ginsburg, Anirban Sen Gupta, Yolanda Cruz.

**Supervision:** Matthew L. Holding, David Ginsburg, Danielle H. Drabeck.

**Validation:** Laura M. Haynes, Jordan A. Shavit.

**Visualization:** Dante Disharoon, Danielle H. Drabeck.

**Writing – original draft:** Dante Disharoon, Danielle H. Drabeck.

**Writing – review & editing:** Matthew L. Holding, Laura M. Haynes, Bipin Chakravarthy Paruchuri, M. Hao Hao Pontius, Krista Golden, Jordan A. Shavit, Karl C. Desch, David Ginsburg, Anirban Sen Gupta, Yolanda Cruz.

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
