## [Decision Letter · Decision Letter 0]

14 Jun 2025

Dear Dr. Drabeck,

Thank you for submitting your manuscript to PLOS ONE. After careful consideration, we feel that it has merit but does not fully meet PLOS ONE’s publication criteria as it currently stands. Therefore, we invite you to submit a revised version of the manuscript that addresses the points raised during the review process.

We look forward to receiving your revised manuscript.

Kind regards,

Karen de Morais-Zani

Academic Editor

PLOS ONE

Journal Requirements:

Authors on this work were supported by the Minnesota IRACDA Program 2K12GM119955 (NIGMS, NIH), the NSF STEM-APWD (DEB 2316784), NIH R01-HL121212, NIH R35-HL171421, and NIH R35-HL150784 (NHLBI).

The authors would like to thank Alexandra Rucavado, Erika Hingst-Zaher, and the Kentucky Reptile Zoo for their help in acquisition and purification of venoms used for this work. Authors on this work were supported by the Minnesota IRACDA Program 2K12GM119955 (NIGMS, NIH), the NSF STEM APWD (DEB 2316784), NIH R01-HL121212, NIH R35-HL171421, and NIH R35-HL150784 (NHLBI). The authors would like to thank Suzanne McGaugh for tremendous guidance as well as lab and resource support. The authors would also like to thank Marvin Neiman (Case Western), Alison Narayan/Sarah Ackenhusen (University of Michigan), and Randy Westrick (Oakland University) for their guidance, feedback, and support.

Authors on this work were supported by the Minnesota IRACDA Program 2K12GM119955 (NIGMS, NIH), the NSF STEM-APWD (DEB 2316784), NIH R01-HL121212, NIH R35-HL171421, and NIH R35-HL150784 (NHLBI).

6. Please amend either the title on the online submission form (via Edit Submission) or the title in the manuscript so that they are identical.

7. Please amend your list of authors on the manuscript to ensure that each author is linked to an affiliation. Authors’ affiliations should reflect the institution where the work was done (if authors moved subsequently, you can also list the new affiliation stating “current affiliation:….” as necessary).

8. Please update your submission to use the PLOS LaTeX template. The template and more information on our requirements for LaTeX submissions can be found at Please%20update%20your%20submission%20to%20use%20the%20PLOS%20LaTeX%20template.%20The%20template%20and%20more%20information%20on%20our%20requirements%20for%20LaTeX%20submissions%20can%20be%20found%20at%20
http://journals.plos.org/plosone/s/latex.

9. We note that Figure 3 and 4 in your submission contain [map/satellite] images which may be copyrighted. All PLOS content is published under the Creative Commons Attribution License (CC BY 4.0), which means that the manuscript, images, and Supporting Information files will be freely available online, and any third party is permitted to access, download, copy, distribute, and use these materials in any way, even commercially, with proper attribution. For these reasons, we cannot publish previously copyrighted maps or satellite images created using proprietary data, such as Google software (Google Maps, Street View, and Earth). For more information, see our copyright guidelines: http://journals.plos.org/plosone/s/licenses-and-copyright.

a. You may seek permission from the original copyright holder of Figure 3 and 4 to publish the content specifically under the CC BY 4.0 license.

Natural Earth (public domain): (http://creativecommons.org/licenses/by/4.0/).%20Please%20be%20aware%20that%20this%20license%20allows%20unrestricted%20use%20and%20distribution,%20even%20commercially,%20by%20third%20parties.%20Please%20reply%20and%20provide%20explicit%20written%20permission%20to%20publish%20XXX%20under%20a%20CC%20BY%20license%20and%20complete%20the%20attached%20form.”%0b%0bPlease%20upload%20the%20completed%20Content%20Permission%20Form%20or%20other%20proof%20of%20granted%20permissions%20as%20an%20%22Other%22%20file%20with%20your%20submission.%0b%0bIn%20the%20figure%20caption%20of%20the%20copyrighted%20figure,%20please%20include%20the%20following%20text:%20“Reprinted%20from%20%5bref%5d%20under%20a%20CC%20BY%20license,%20with%20permission%20from%20%5bname%20of%20publisher%5d,%20original%20copyright%20%5boriginal%20copyright%20year%5d.”%0b%0bb.%20If%20you%20are%20unable%20to%20ob" http://www.naturalearthdata.com/

10. Please include captions for your Supporting Information files at the end of your manuscript, and update any in-text citations to match accordingly. Please see our Supporting Information guidelines for more information: http://journals.plos.org/plosone/s/supporting-information.

Reviewers' comments:

Reviewer's Responses to Questions

**Comments to the Author**

1. Is the manuscript technically sound, and do the data support the conclusions?

Reviewer #1: Partly

Reviewer #2: Yes

2. Has the statistical analysis been performed appropriately and rigorously?

Reviewer #1: I Don't Know

Reviewer #2: Yes

3. Have the authors made all data underlying the findings in their manuscript fully available?

Reviewer #1: Yes

Reviewer #2: Yes

4. Is the manuscript presented in an intelligible fashion and written in standard English?

Reviewer #1: Yes

Reviewer #2: Yes

Reviewer #1: In this manuscript, Matthew Holding and colleagues investigated the venom resistance of M. domestica. In different experiments, they examined whether the decreased binding between VWF and botrocetin results in functional resistance to coagulopathic effects, and whether M. domestica is capable of neutralizing venom components beyond CTLs.

I have several questions and remarks on the current version of this manuscript:

-Overall, it is nice data but the focus on VWF (as was expected from the title and abstract) is quickly lost, going to general resistance to coagulopathic effects. Therefore, I would suggest to change the title. In addition, often more explanation is needed at the start of a results paragraph, to explain why a certain experiment is performed.

For figure 1 and 2: “As expected, … platelet accumulation on the microfluidic surface was strongly inhibited by botrocetin and convulxin” (L135-136) Here, a short text is needed to explain why this should be expected. This is provided in the material and methods section so it could be removed there from both ‘Microfluidic Assays’ and ‘Aggregation Assays’ paragraphs, and be added to results section.

For figure 4, it is now not clear how this figure adds to the story (especially not with the current title/abstract), as the main research question here focusses on VWF, but in Figure 4 suddenly metalloproteinase inhibition is tested. Also, suddenly different venoms are used compared to Figure 3. The rationale for this should be provided in the results section.

-Figure 1: the authors conclude that opossum VWF requires a greater shear force to elongate. Is this significant? This is not indicated in the graph.

-Supplemental figure 1: from this figure, I’m not convinced with the conclusion that ‘there were no obvious differences in multimer distribution’. First, quantification (% to total) of high, intermediate and low molecular weight multimers is needed. Second, an equal concentration should be loaded on the gel to be able to compare, which seems not to be the case here.

-In Figure 2, images of fibrin formation under flow would be of added value.

-The human control is lacking in Figure 3, while it was nicely added in Figure 1 and 2. Regarding the lack of this comparison, the conclusion on L277-279 (Aspercetin appears to be able to function nearly as well in opossums as it does in humans in microfluidics assays (in both cases it has weak function), but not in whole blood assays) can’t be drawn as aspercetin was not tested here in whole blood assays using human whole blood.

-In the graphical abstract, aspercetin is mentioned as ‘not resistant’ in platelet adhesion, while convulxin and botrocetin are mentioned as ‘moderately resistant’. This is not correct when looking at Figure 2J-K?

Minor:

-The abbreviations CTL and SVMP are used in the introduction, however not written out fully when used for the first time. Please adjust this.

-“Concordant with the whole blood aggregometry analyses” (L145< L150) is mentioned in the results section of Figure 2. However, only microfluidic experiments using PRP are described thus far, and whole blood aggregometry experiments follow in Figure 3. This text should be adjusted.

-In the results section, refer to the specific subfigures (e.g. Figure 3B instead of Figure 3).

-Typo in the graphical abstract: sheer stress should be shear stress. Also, the shear stress mentioned here is 80 dyn/cm2 while 60 dyn/cm2 is mentioned in the figure legends.

Reviewer #2: This article discusses the resistance to envenomation of a small and poorly studied species of Opossum, focusing on coagulopathic effects. The article provides robust information by covering different types of coagulopathic activities to better understand the effects of resistance to envenomation, as well as using different species and isolated toxins, considering the animal ecology and its possible targets of resistance to envenomation. Therefore, I recommend the publication of the article, which brings new information about prey-predator coevolution (Arms race), contributing scientifically to a better understanding of the phenomenon.

Minor suggestions:

Please include an item in the Methodology section to describe statistical analysis used on the article

Line 59 – please include the description of CTLs here – once it´s the first time that the abbreviation is described

Line 300 – verifiy the reference

Sugegstion on figure 1: Do the AUC curves to statistically compare between Human and oposum shear stress could enrich the results.

**Do you want your identity to be public for this peer review?** For information about this choice, including consent withdrawal, please see our Privacy Policy

Reviewer #1: No

Reviewer #2: **Yes: ** Caroline Fabri Bittencourt Rodrigues

---

## [Author Response · Author response to Decision Letter 1]

15 Aug 2025

Please see the attached "Response to Reviewers" pdf document for a more readable format of the responses below.

Response to reviews (our responses in bold text):

We thank both reviewers for their thoughtful and constructive feedback on our manuscript. We were pleased to see enthusiasm for the dataset and overall study design, and we have revised the manuscript to improve clarity, strengthen the rationale for experimental progression, and more clearly connect our results to the role of von Willebrand Factor (VWF). In particular, we added new contextual framing throughout the Introduction and Results section, clarified figure references, corrected terminology, and incorporated a dedicated Statistical Analysis section in response to reviewer suggestions. Additionally, we have reworked much of the Discussion section to both shorten its length and explicitly connect the discussed points to the updated framing language in the Introduction and Results. Below, we address each comment in detail, indicating the specific changes made to the text and our rationale where appropriate. We appreciate the reviewers’ engagement and believe the manuscript is now stronger and more accessible as a result.

Reviewer #1: In this manuscript, Matthew Holding and colleagues investigated the venom resistance of M. domestica. In different experiments, they examined whether the decreased binding between VWF and botrocetin results in functional resistance to coagulopathic effects, and whether M. domestica is capable of neutralizing venom components beyond CTLs.

I have several questions and remarks on the current version of this manuscript:

-Overall, it is nice data but the focus on VWF (as was expected from the title and abstract) is quickly lost, going to general resistance to coagulopathic effects. Therefore, I would suggest to change the title.

We appreciate the need for clarity of message in our title. The inclusion of VWF in the title emphasizes the key role of VWF in platelet and fibrin deposition as the known major target of botrocetin, as well as the novel finding of the behavior of opossum VWF under shear. However, we see the reviewer’s point that the prior title placed too much connection between the nature of the resistance assays and VWF specifically. We have changed the title to the following: “Comparative microfluidic and enzymatic analyses reveal multifaceted snake venom resistance and novel VWF behavior in the opossum Monodelphis domestica”

While VWF plays a central mechanistic role in venom resistance, we intentionally investigated downstream physiological responses, including resistance to SVMPs, to demonstrate the integrated and systemic nature of the phenotype and to place investigation of species-specific adaptive venom resistance within the same methodological context used in most other studies of resistance to viper venoms (that of inhibiting hemorrhagic SVMPs). This broader systems-level framing aligns with our title and is essential for establishing organism-level functional resistance. We address this integration of vWF mediated resistance with multifaceted resistance in the introduction on lines 103-105:

“These findings suggest that small-bodied species share venom SNACLEC resistance, but it remains unclear whether this reflects integrated, physiological resistance to snake venom that includes resistance to multiple protein classes.”

In addition, often more explanation is needed at the start of a results paragraph, to explain why a certain experiment is performed.

We appreciate the reviewer’s suggestion and have revised the manuscript in several places to improve the clarity and logical flow of the Introductory material and into the Results section. Specifically, the final paragraph of the introduction explicitly includes hypotheses and associated data-centric predictions. Then, we added brief explanatory sentences at the beginning of each results paragraph to clearly articulate the rationale behind each experimental step. Additionally, we have made changes to the introduction section that make the justification for each aspect more clear and concise, further justify the metalloproteinase activity tests, and differentiate the VWF elongation results from the rest of the manuscript. The Results section changes are incorporated at the following locations:

Line 344 – Comparative microfluidic analysis of VWF behavior

Addition: To determine whether evolved sites in M. domestica VWF [as described in 10] result in functional resistance under physiologically relevant conditions, we first assessed VWF elongation, platelet adhesion, and fibrin formation using microfluidic assays that simulate in vivo shear forces

Line 360 – Prior to platelet adhesion results

Addition: We next compared the platelet adhesion response of opossum and human PRP to purified venom SNACLECs to evaluate whether resistance is preserved in integrated clotting dynamics under flow

Line 415 – Prior to fibrin kinetics paragraph

Addition: Due to the complex nature of blood coagulation, we quantified not only the effects of SNACLECs on platelet adhesion but also fibrin deposition so as to evaluate downstream coagulopathic disruptions.

Line 441 – Whole Blood Aggregation

Addition: To further assess species-specific resistance to venom in the milieu of whole blood, we used whole blood aggregometry to test the platelet aggregation response of M. domestica to both whole venom and purified SNACLECs.

Line 508 – Venom Metalloproteinase Inhibition

Addition: To evaluate whether resistance in M. domestica extends beyond VWF-targeting SNACLECs and to test for ecological specificity of serum-based SVMP inhibition, we conducted a complete reciprocal cross of 3 viperid species venoms with 3 mammal serums. Venoms were selected to represent 1) a sympatric species (B. jararaca) for which ecological and other data predict opossum resistance, 2) an allopatric species (C. oreganus) for which another mammal (ground squirrels) have evolved venom resistance 3) an allopatric distantly related viper (B. arietans) (see Fig 4 top and middle panels.)

For figure 1 and 2: “As expected, … platelet accumulation on the microfluidic surface was strongly inhibited by botrocetin and convulxin” (L135-136) Here, a short text is needed to explain why this should be expected. This is provided in the material and methods section so it could be removed there from both ‘Microfluidic Assays’ and ‘Aggregation Assays’ paragraphs, and be added to results section.

We thank the reviewer for this helpful suggestion. We have revised the opening sentence of this paragraph in the Results section (now line 360-365) to include a brief explanation of why botrocetin and convulxin were expected to inhibit platelet adhesion in human blood. Specifically, we now state that this is based on prior work demonstrating the strong disruptive effect of these CTLs on platelet function. This clarifies the basis of the expectation for readers unfamiliar with the specific toxins. We also streamlined redundant text in the Methods to avoid repetition.

Updated Text (Line 360-361):

“Control experiments with human PRP showed that platelet accumulation on the microfluidic surface was strongly inhibited by botrocetin and convulxin, and weakly inhibited by aspercetin (Fig 2A-D), which is consistent with prior work demonstrating that botrocetin and convulxin strongly activate or disrupt platelet function in human blood [37,38].”

For figure 4, it is now not clear how this figure adds to the story (especially not with the current title/abstract), as the main research question here focusses on VWF, but in Figure 4 suddenly metalloproteinase inhibition is tested. Also, suddenly different venoms are used compared to Figure 3. The rationale for this should be provided in the results section.

We agree that additional context was needed. We have added specific introduction section material that both justifies and prepares the reader to encounter SVMP assays:

Lines 92-95: “This mechanism is further exacerbated by snake venom metalloproteinases (SVMPs), that cleave plasma proteins and break down vascular integrity to release other venom compoents like SNACLECs into circulation, further compounding coagulopathic damage [19,20].”

Lines 138-140: “We extend these tests of species specificity through assays of serum inhibition of the SVMPs to investigate the potential for adaptive, integrated resistance to multiple venom protein classes.”

We also added a transitional paragraph to the beginning of the Results subsection presenting Figure 4 (line 508-514) to explain how this assay complements the SNACLEC/VWF-focused experiments. Specifically, we clarify that metalloproteinase inhibition was tested to determine whether Monodelphis domestica exhibits broader resistance to venom effects beyond VWF-SNACLEC interactions. We also clarify the ecological rationale for the venom selection, which includes sympatric, partially sympatric, and allopatric species to assess the specificity of serum-based inhibition.

“To evaluate whether resistance in M. domestica extends beyond VWF-targeting SNACLECs and to test for ecological specificity of serum-based SVMP inhibition, we conducted a complete reciprocal cross of 3 viperid species venoms with 3 mammal serums. Venoms were selected to represent 1) a sympatric species (B. jararaca) for which ecological and other data predict opossum resistance, 2) an allopatric species (C. oreganus) for which another mammal (ground squirrels) have evolved venom resistance 3) an allopatric distantly related viper (B. arietans) (see Fig 4 top and middle panels.)”

-Figure 1: the authors conclude that opossum VWF requires a greater shear force to elongate. Is this significant? This is not indicated in the graph.

Thank you for this important request. The difference between the percent elongation of human and opossum VWF is significant at 40 and 60 dyn/cm2 with p values of 0.021 and 0.027, respectively. We have edited Figure 1 by adding asterisks to denote statistical significance. Additionally, the text and Figure 1 caption have been updated to reflect this information.

In Figure 1 caption: “Asterisks denote statistically significant differences in percent elongation (-value <0.05.”

In the text: “Compared to human VWF, opossum VWF required an additional 20 dvn/cm2 to fully elongate, and demonstrated statistically reduced elongation at shear stresses of 40 and 60 dyn/cm2.” (lines 354-356)

-Supplemental figure 1: from this figure, I’m not convinced with the conclusion that ‘there were no obvious differences in multimer distribution’. First, quantification (% to total) of high, intermediate and low molecular weight multimers is needed. Second, an equal concentration should be loaded on the gel to be able to compare, which seems not to be the case here.

We appreciate the reviewer’s attention to detail and agree that quantitative comparison of multimer distribution should include both densitometric analysis and equal loading. We have moderated the language here to reflect both the contents of the gel image and our original goal in conducting this multimer analysis, that of confirming the existence of high molecular weight multimers in this distantly related mammalian species (lines 349-356).

-In Figure 2, images of fibrin formation under flow would be of added value.

Thank you, and we agree. We have edited Figure 2 to display both platelet and fibrin fluorescence. Additionally, Supplementary Video 1 has been added so that the kinetics of platelet adhesion and fibrin generation under flow can be better appreciated.

The human control is lacking in Figure 3, while it was nicely added in Figure 1 and 2. Regarding the lack of this comparison, the conclusion on L277-279 (Aspercetin appears to be able to function nearly as well in opossums as it does in humans in microfluidics assays (in both cases it has weak function), but not in whole blood assays) can’t be drawn as aspercetin was not tested here in whole blood assays using human whole blood.

We appreciate the reviewer’s concern. Due to the limited availability of purified CTLs such as aspercetin, convulxin, and botrocetin—some of which are rare, previously fractionated venoms—our study prioritized using these reagents in assays that would generate novel comparative data, specifically in Monodelphis domestica whole blood. Human responses to all three CTLs in PRP and whole blood have been extensively characterized in prior studies (e.g., Drabeck et al. 2020; Rucavado et al. 2001), including using the same purified fractions employed here. To address this comment, we have revised the manuscript text to remove new inferences about human whole blood responses and now cite the relevant prior literature more clearly.

Revised text:

Line 480-484: “Due to limited quantities of these purified SNACLECs, we prioritized their use in Monodelphis domestica whole blood. Human platelet responses to these same protein fractions have been previously characterized in PRP [9,17], and we refer readers to those studies for comparative context.”

Line 595-596: “Aspercetin appears to be able to function nearly as well in opossums as it does in humans under flow (in both cases it has weak function), but not in whole blood or previous PRP assays [10], highlighting the importance of integrated assays that incorporate the complex milieu of the blood coagulation system as well the dynamic shear forces that act upon [17,39,46].”

In the graphical abstract, aspercetin is mentioned as ‘not resistant’ in platelet adhesion, while convulxin and botrocetin are mentioned as ‘moderately resistant’. This is not correct when looking at Figure 2J-K?

We thank the reviewer for their careful attention. As shown in Figure 2K, opossum platelet adhesion is statistically significantly preserved in the presence of botrocetin relative to human PRP, reflecting resistance. In contrast, the response to aspercetin in opossum PRP is nearly identical to that in human PRP—indicating a lack of resistance. Convulxin, while severely impairing adhesion in both species, allowed a modest degree of residual adhesion in opossum PRP which we were classifying as resistance. However, we have appropriately changed this interpretation to “Not Resistant” based on the significant thresholds.

Minor:

-The abbreviations CTL and SVMP are used in the introduction, however not written out fully when used for the first time. Please adjust this.

We have checked the manuscript throughout to ensure initial definition (lines 84 and 93) and proper continued usage of these abbreviations. Additionally, following conversations with a colleague that studies these proteins, we have switched to the abbreviation SNACLEC for the venom c-type lectin-like proteins, as this is the more common usage in the field for differentiation botrocetin and related proteins from the true venom CTLs, which are located on a different chromosome.

“Concordant with the whole blood aggregometry analyses” (L145< L150) is mentioned in the results section of Figure 2. However, only microfluidic experiments using PRP are described thus far, and whole blood aggregometry experiments follow in Figure 3. This text should be adjusted.

We agree with the reviewer and appreciate the clarification. To avoid referencing data that the reader has not yet encountered, we have revised the section to remove comparison with results that are not yet presented:

Lines 373-381: “The effect of botrocetin on platelet adherence to the microchannel was decreased in M. domestica (Fig 2 B, F, J): botrocetin caused a more severe defect in the total extent of platelet adhesion than aspercetin in human PRP, but a less severe defect than aspercetin in opossum PRP (Fig 2I-K). Aspercetin and convulxin caused a similar response in both species (Fig 2K). The effect of botrocetin on platelet adherence to the microchannel was also significantly decreased in M. domestica when compared to the same in human (Fig 2 B & F, K). In human PRP, botrocetin caused a 66 ± 16% reduction in the overall extent of platelet adhesion compared to the negative control, whereas it only caused a 39 ± 15% reduction in opossum PRP (Fig 2K”

-In the results section, refer to the specific subfigures (e.g. Figure 3B

---

## [Decision Letter · Decision Letter 1]

28 Aug 2025

Dear Dr. Holding,

We look forward to receiving your revised manuscript.

Kind regards,

Karen de Morais-Zani

Academic Editor

PLOS ONE

Journal Requirements:

Additional Editor Comments:

One of the reviewers suggests some minor corrections. We look forward to receiving your revised manuscript.

Reviewers' comments:

Reviewer's Responses to Questions

**Comments to the Author**

Reviewer #1: All comments have been addressed

Reviewer #2: All comments have been addressed

2. Is the manuscript technically sound, and do the data support the conclusions?

Reviewer #1: Yes

Reviewer #2: Yes

3. Has the statistical analysis been performed appropriately and rigorously?

Reviewer #1: Yes

Reviewer #2: Yes

4. Have the authors made all data underlying the findings in their manuscript fully available?

Reviewer #1: Yes

Reviewer #2: Yes

5. Is the manuscript presented in an intelligible fashion and written in standard English?

Reviewer #1: Yes

Reviewer #2: Yes

Reviewer #1: The authors have adequately and extensively addressed my comments. I have no further questions or comments.

Reviewer #2: The authors have satisfactorily addressed the issues raised in the first review. Only a few editing issues should be considered for submission.

Line 75 - different font from the rest of the text

on line 419, replace the term C-type lectins for SNACLECs.

line 455: italic missing on B. jararaca

line 557: missing parentesis

references with a different font from the rest of the text

duplicated references ( 49 to 51)

**Do you want your identity to be public for this peer review?** For information about this choice, including consent withdrawal, please see our Privacy Policy

Reviewer #1: No

Reviewer #2: **Yes: ** Caroline Fabri Bittencourt Rodrigues

---

## [Author Response · Author response to Decision Letter 2]

2 Sep 2025

Reviewer #1: The authors have adequately and extensively addressed my comments. I have no further questions or comments.

We thank the reviewer for their previous comments and for look reviewing our changes.

Reviewer #2: The authors have satisfactorily addressed the issues raised in the first review. Only a few editing issues should be considered for submission.

We appreciate the reviewer’s attention to these additional details and have implemented all suggested changes.

Line 75 - different font from the rest of the text

Font changed to Roboto

on line 419, replace the term C-type lectins for SNACLECs.

SNACLECs added.

line 455: italic missing on B. jararaca

“B. jararaca” has been italicized.

line 557: missing parentesis

Parentheses added

references with a different font from the rest of the text

References font changed to Roboto

duplicated references ( 49 to 51)

We have removed the duplicated references.

---

## [Editor Report · Decision Letter 2]

4 Sep 2025

Comparative microfluidic and enzymatic analyses reveal multifaceted snake venom resistance and novel VWF behavior in the opossum Monodelphis domestica

PONE-D-25-27609R2

Dear Dr. Holding,

We’re pleased to inform you that your manuscript has been judged scientifically suitable for publication and will be formally accepted for publication once it meets all outstanding technical requirements.

Kind regards,

Karen de Morais-Zani

Academic Editor

PLOS ONE
---

## [Editor Report · Acceptance letter]

PONE-D-25-27609R2

PLOS ONE

Dear Dr. Holding,

I'm pleased to inform you that your manuscript has been deemed suitable for publication in PLOS ONE. Congratulations! Your manuscript is now being handed over to our production team.

Kind regards,

on behalf of

Dr. Karen de Morais-Zani

Academic Editor

PLOS ONE